# The Effect of Three Mediterranean Diets on Remnant Cholesterol and Non-Alcoholic Fatty Liver Disease: A Secondary Analysis

**DOI:** 10.3390/nu12061674

**Published:** 2020-06-04

**Authors:** Angelo Campanella, Palma A. Iacovazzi, Giovanni Misciagna, Caterina Bonfiglio, Antonella Mirizzi, Isabella Franco, Antonella Bianco, Paolo Sorino, Maria G. Caruso, Anna M. Cisternino, Claudia Buongiorno, Rosalba Liuzzi, Alberto R. Osella

**Affiliations:** 1Laboratory of Epidemiology and Biostatistics, National Institute of Gastroenterology, “S. de Bellis” Research Hospital, Castellana Grotte (Bari), Via Turi 27, 70013 Castellana Grotte, Italy; angelocampanella7@gmail.com (A.C.); catia.bonfiglio@irccsdebellis.it (C.B.); antonella.mirizzi@irccsdebellis.it (A.M.); isabella.franco@irccsdebellis.it (I.F.); antonella.bianco@irccsdebellis.it (A.B.); paolosorino96@libero.it (P.S.); buongiorno.claudia@gmail.com (C.B.); liuzzirosalba@libero.it (R.L.); 2Department of Clinical Pathology, National Institute of Gastroenterology, “S. de Bellis” Research Hospital, Castellana Grotte (Bari), Via Turi 27, 70013 Castellana Grotte, Italy; mina.iacovazzi@irccsdebellis.it; 3Scientific and Ethical Committee, University Hospital Policlinico, 70124 Bari, Italy; gmisciag@libero.it; 4Ambulatory of Clinical Nutrition, National Institute of Gastroenterology, “S. de Bellis” Research Hospital, Castellana Grotte (Bari), Via Turi 27, 70013 Castellana Grotte, Italy; gabriella.caruso@irccsdebellis.it (M.G.C.); annamaria.cisternino@irccsdebellis.it (A.M.C.)

**Keywords:** fasting REM-C levels, NAFLD severity score, insulin resistance

## Abstract

Background: Elevated fasting remnant cholesterol (REM-C) levels have been associated with an increased cardiovascular risk in patients with metabolic syndrome (Mets) and Non-Alcoholic Fatty Liver Disease (NAFLD). We aimed to estimate the effect of different diets on REM-C levels in patients with MetS, as well as the association between NAFLD and REM-C. Methods: This is a secondary analysis of the MEDIDIET study, a parallel-arm Randomized Clinical Trial (RCT). We examined 237 people with MetS who underwent Liver Ultrasound (LUS) to assess the NAFLD score at baseline, 3-, and 6-months follow-up. Subjects were randomly assigned to the Mediterranean diet (MD), Low Glycemic Index diet (LGID), or Low Glycemic Index Mediterranean diet (LGIMD). REM-C was calculated as [total cholesterol–low density lipoprotein cholesterol (LDL-C)–high density lipoprotein cholesterol (HDL-C)]. Results: REM-C levels were higher in subjects with moderate or severe NAFLD than in mild or absent ones. All diets had a direct effect in lowering the levels of REM-C after 3 and 6 months of intervention. In adherents subjects, this effect was stronger among LGIMD as compared to the control group. There was also a significant increase in REM-C levels among Severe NAFLD subjects at 3 months and a decrease at 6 months. Conclusions: fasting REM-C level is independently associated with the grade of severity of NAFLD. LGIMD adherence directly reduced the fasting REM-C in patients with MetS.

## 1. Introduction

Hyperlipidemia is a condition characterized by elevated levels of serum lipids. Excess lipids in the blood commonly accumulate in the walls of arteries and several studies have established the central role of hyperlipidemia in the development of cardiovascular diseases (CVD) [1,2]. Other important studies have irrefutably demonstrated a strong relationship between hypercholesterolemia, mortality, and the incidence of CVD [3,4]. In particular, high Low-Density Lipoprotein levels (LDL-C), in combination with low High-Density Lipoprotein levels (HDL-C), are associated with an increased risk of atherosclerosis and ischemic heart disease.

Preventive measures, pharmacological therapy, and lifestyle changes have been focused on reducing LDL-C levels, which is currently the main target of lipid-lowering therapies [5,6,7]. However, even after a reduction of LDL-C to the recommended concentrations, there is still a considerable residual risk of recurrent cardiovascular events, so lowering LDL-C levels alone is insufficient [8]. Some of this residual risk may be attributed to elevated remnant cholesterol (REM-C) levels [9].

REM-C is the cholesterol content of triglyceride-rich lipoproteins, which in the fasting state include the Very Low-Density Lipoproteins (VLDL) and the Intermediate-Density Lipoproteins (IDL) produced by the liver. In the non-fasting state, REM-C also includes chylomicron remnants [10,11]. High levels of non-fasting plasma triglycerides are a marker of elevated non-fasting REM-C [10,12,13] and are associated with an increased risk of cardiovascular disease [10,12,13]. Since triglycerides per se are unlikely to have a direct role in provoking cardiovascular disease [12,13,14,15,16], REM-C is more likely to be the causal factor.

Irawati et al. (2016) [17] have shown a substantial accumulation of pro-atherogenic REM-C in subjects with metabolic syndrome (MetS), who are well known to be at risk of developing CVD. Increased plasma triglycerides and decreased HDL-C levels are typical components of MetS. Another condition associated with MetS is Non-Alcoholic Fatty Liver Disease (NAFLD), a potential independent CVD risk factor which leads to increases in liver fat, inflammation, and oxidative stress [18,19]. Indeed, the early manifestation of NAFLD is triglyceride accumulations in the liver associated with insulin resistance (IR) [20]. In patients with NAFLD, higher REM-C values were associated with greater liver disease severity and REM-C levels were found to be predictive of CVD [21].

Lifestyle changes, including dietary modifications (reducing carbohydrate and fat intake), physical activity, reducing alcohol intake, and weight loss, are the mainstay in the management of hypercholesterolemia and hypertriglyceridemia [22,23,24]. Several studies have been focused on the relationship between MetS and the Mediterranean diet, which features significant amounts of fibers, antioxidants, vegetable proteins, polyunsaturated, and monounsaturated fats. It has also been suggested that this way of eating decreases the risk of CVD and improves NAFLD [25]. These interventions have shown that the Mediterranean diet can reduce plasma triglyceride levels by up to 60% [22]. In adults, the Mediterranean Diet has also been demonstrated to be effective in reducing the risk of MetS [26,27].

In our Institution, a Randomized Clinical Trial (RCT) was conducted to estimate the effect of different diets on the MetS score and on its components after a six-month intervention period [28]. As no studies of the impact of diet on the relationship between REM-C and MetS have yet been reported, we aimed to estimate the effect of different diets on REM-C levels in patients with MetS, as well as the association between NAFLD and REM-C in patients enrolled in the RCT, by performing a secondary analysis [29].

## 2. Materials and Methods

### 2.1. Study Design

The original RCT is registered at https://clinicaltrials.gov/, Identifier: NCT02356952 [30].

MEDIDIET was a parallel-arm RCT. Subjects with MetS were selected from the MICOL cohort study conducted at the National Institute of Gastroenterology “Saverio de Bellis” Research Hospital [31,32]. MICOL is an ongoing population study, which was started in 1985, including subjects randomly sampled (aged 30–89 years) from the electoral rolls of Castellana Grotte, a town in southern Italy (Apulia region). The cohort was followed up in 1992–1993 and in 2005–2006. In total, 2472 men and women were enrolled. In the second follow-up, the cohort was fed with 1273 randomly sampled young participants (30–49 years old) to compensate for aging of the cohort.

### 2.2. Participants Selection

Between December 2007 and April 2008, MICOL subjects who had been screened in 2005–2006 and had MetS (1042 subjects) were invited to undergo further examination: 556 subjects responded and 163 of 387 subjects (100 males, 63 females) were still affected by MetS. We included subjects treated with statins, anti-hypertensives, and oral antidiabetics, but excluded subjects in insulin treatment. Patients were requested not to change their exercise habits after enrollment in the study.

The trial was conducted in collaboration with General Practitioners, approved by the Ethics Committee of our Institution (D.Legs 502/92, Conv. N.54-2 marzo 2006) in accordance with the Helsinki Declaration, and all participants provided written informed consent.

### 2.3. Randomization

Participants were randomly assigned by simple randomization procedures (computerized random numbers sequence) to one of three diets or the Control group (C); a one-to-one ratio was used to allocate the subjects.

Blinding was maintained, firstly assuring the staff and participants that each diet was based on healthy principles. Participants were followed for the duration of the trial and the dietitian was assigned on a daily random basis. Moreover, only one intervention group was called in each day to reduce to a minimum the information exchange among participants. Staff members who assessed the outcomes were unaware of the diet assigned. Only one of two radiologists performed outcome measurements each day and this order was also randomly assigned. In the follow-up performed at the third and sixth months, the radiologists were unaware of the previous measurements.

### 2.4. Baseline Examination

Initial screening included a complete medical history and physical examination. Participants were asked to complete a structured questionnaire administered by a trained interviewer. Blood pressure at rest was measured by a trained nurse, using a sphygmomanometer with the appropriate cuff. Blood samples were drawn in the morning after overnight fasting and biochemical measurements were performed using standardized methods in the central laboratory of our institution. The serum was separated into two different aliquots. One aliquot was immediately stored at −80 °C. Anthropometric measurements (weight, height, waist circumference) were taken by three dietitians. Weight measurements were taken using SECA mechanical scales (Model 700; SECA, Hamburg, Germany), while height was measured using a wall-mounted altimeter (Model 206; 220 cm; SECA, Hamburg, Germany). Dietitians also administered a validated semi-quantitative food frequency questionnaire [33] and carried out Bio-Impedentiometric Analysis (BIA) (Akern SRL, Pontassieve FI, Italy). Visceral Adiposity Thickness (VAT) was measured with Ultrasound scans using a strict protocol. It was defined as the distance between the anterior wall of the aorta and the internal face of the rectus abdominis muscle perpendicular to the aorta [34].

### 2.5. Dietary Intervention

The recommended diets were provided in brochure format, with graphical explanations organized according to a traffic light system: with a list of foods that can be consumed frequently (green foods), sometimes (yellow foods), and never (red foods). The brochure also contained a dietary record, where participants indicated daily the code of each food consumed at breakfast, lunch, dinner, and during snack time. There were no calorie restrictions. Diets composition is shown in Appendix A.

We studied the effects of the following diets: Mediterranean diet (MD), built using the Trichopoulou A. et al. study [35]; Low Glycemic Index diet (LGID) based on the Elia A. study [36], and Low Glycemic Index Mediterranean diet (LGIMD), created by integrating the Trichopolou A. et al. [35] and Elia A. studies [36] and adapting them to our population.

Individuals recorded what they ate on a daily diet diary. The main objectives pursued in the creation of the diets and the administration and monitoring tools were: (1) to let subjects choose their foods and (2) to help them monitor what they ate. The characteristics of the three diets and their main nutritional composition are described in Appendix A and Appendix B. Mean intake of Energy, Alcohol, and Macronutrients were calculated using Metadieta^®^ software, version 3.7. Patients were asked not to change their physical activity after enrollment in the study.

Control group participants were only given general nutritional counseling and were advised to maintain their lifestyle. The control group was followed up with the same methodology.

### 2.6. Outcomes

In the protocol, primary outcome measures were MetS, MetS score, and its components. Secondary outcomes were anthropometric measurements and biochemical markers and Non-Alcoholic Fatty Liver Disease (NAFLD) score (measured by Liver Ultrasound (LUS)).

The quantitative serum levels of Total Cholesterol (TC), Triglycerides (TGL), Glucose (G), HDL-Cholesterol (HDL-C), and Insulin (INS) were evaluated using the UniCelDxC and DxI 660i Integrated System (Beckman Coulter, Fullerton, CA, USA). The timed endpoint method was used for chemicals and the chemiluminescent method for insulin. LDL-C was assessed using Friedewald’s formula [37].

Insulin resistance and beta-cell function were estimated by the Homeostasis Model Assessment (HOMA) Index = (glucose × insulin)/22.5 [38].

In accordance with the reference values routinely adopted in our laboratory, we considered out-of-range values as: total cholesterol > 5.18 mmol/L, triglycerides > 1.86mmol/L, glucose > 6.94 mmol/L, HDL-C < 1.04 mmol/L, insulin > 29 μIU/mL, LDL-C > 3.88 mmol/L, and HOMA > 2.5.

REM-C was calculated as [total cholesterol−LDL-C−HDL-C] and we considered a value >30 mg/dL as altered, in accordance with the literature for fasting REM-C [39].

### 2.7. Implementation

Subjects were followed up monthly for dietary counseling, to check diaries, and to control anthropometric parameters. After 12 and 24 weeks from the beginning of the study, the subjects again underwent fasting blood sampling and BIA.

### 2.8. Statistical Analysis

The primary analysis was intention-to-treat. Data were extensively described by means of Tables by using Mean (±SD) and frequency (%) as descriptive measures for continuous and categorical variables, respectively.

The Mediterranean Adequacy Index (MAI) [40,41] was calculated to assess the compliance with the prescribed diet. Random week and week-end days were chosen from the second and fourth month of intervention. The MAI was estimated according to gender and month to clearly describe compliance. Compliance was defined as positive if the subject’s ratio of percentage of calories derived from LGID or MD or LGIMD foods, divided by percentage of calories from foods not in the LGID, MD or LGIMD, was equal to or above the median value for the whole population under study. A median value of 7.5 with an inter-quantile range (IQR) of 5.4 was expected, as established by the reference Italian Mediterranean Diet [40]. An Analysis of Variance for repeated measures was performed to analyze the differences among group means (Diet by Time). This statistical technique was chosen because it is computationally elegant and relatively robust against violations of its assumptions, provides strong (multiple sample comparison) statistical analysis, and it has been adapted to the analysis of a variety of experimental designs. As REM-C was not normally distributed, a natural logarithmic transformation was applied. A post estimation procedure was then applied to contrast the estimated means by taking a reference category, in this case Control group at baseline. Margins statistics were calculated from predictions of a previously fit Analysis of Variance model and displayed graphically.

Statistical analysis was performed using Stata statistical software (version 16.1), StataCorp, 4905 Lakeway Drive, College Station, TX, USA (the Stata code is available upon request from the email: arosella@irccsdebellis.it).

## 3. Results

### Sample Description

Participant characteristics are shown in Table 1. At Baseline, there was no significant difference for any of the variables considered among C and the intervention groups (LGID, MD, and LGIMD). There were 61, 62, 57, and 57 subjects in the C, LGID, MD, and LGIMD groups, respectively. About 40% were females and the mean age was 57.6 (11.8) (men 56.8 (12.0), women 58.8 (11.3)). Mean age was 54.9 (13.9) for controls, 57.5 (10.7) for LGID, 59.4 (10.4) for MD, and 58.3 (9.8) for LGIMD. Baseline REM-C levels were slightly different among groups, but these differences did not reach statistical significance. Systolic Blood Pressure (SBP) and Diastolic Blood Pressure (DBP) were higher than the normal range. As expected, all subjects were obese with a BMI of about 33.0 for most groups and 34.93 for LIGMD. Biochemical markers other than REM-C were equally distributed among the control and intervention groups.

Overall, NAFLD prevalence in the sample was 82.28% and it was more prevalent in men (88.89%) than in women (72.04%). NAFLD severity grade was Absent (17.72%), Mild (18.57%), Moderate (41.77%), and Severe (21.94%). Men had more severe grades of NAFLD.

Table 2 shows descriptive statistics for NAFLD severity by Intervention group and Time. At baseline, in NAFLD absent subjects, REM-C in the control group was lower than in subjects who underwent dietary intervention, but this difference did not reach statistical significance. In all other categories of NAFLD, REM-C were equally distributed.

Overall, adherence to the prescribed diet was 72.82%. The adherence was higher in women (79.98%) than in men (68.85%). Figure 1 shows the change over time of REM-C level by degree of NAFLD severity and type of intervention, both for the whole sample and for the adherent subjects. In the whole sample, there was a decreasing trend in REM-C levels in Absent or Mild NAFLD subjects for MD and LGIMD diets. The same trend was observed in subjects with moderate or severe NAFLD randomized to the MD diet. Instead, in adherent subjects, a decreasing trend was observed with the LGIMD diet for moderate or severe NAFLD.

Results from Repeated Measures Analysis of Variance for Diet and Time are shown in Table 3. In the whole sample, there were statistically significant principal effects for MD (−23.76, 95%CI −43.97; −3.56) and LGIMD (−36.40, 95%CI −56.61; −16.20). Principal effects of Time were of small size and were not statistically significant. Statistically significant principal effects of Diets (increasing levels of REM-C) and Time were observed in adherent subjects. It is worth noting that among Adherent subjects, the referent value is the LGID.

As regards the test of linear hypothesis, contrast of Time within each level of Diet evidenced a statistically significant effect for any contrast of Time (Three and Six months vs. Baseline) and Diet, except for Control Diet. These contrasts behaved in a similar way for both the whole sample and adherent subjects. These results are graphically displayed in Figure 2.

Results from Repeated Measures Analysis of Variance for NAFLD Severity and Time are shown in Table 4. In the whole sample, there were no statistically significant principal effects for NAFLD severity, but there were statistically significant effects for Time. There was a statistically significant principal effect of NAFLD severity, but not a principal effect of Time.

As regards the test of linear hypothesis, contrast of Time within each level of NAFLD severity evidenced a statistically significant effect for Time (three and six months) when NAFLD was Absent and for Moderate NAFLD at three months when the whole sample was considered. For Adherent Subjects, the contrast was statistically significant, but in an opposite way. For Moderate NAFLD, REM-C levels decreased at three and six months. Instead, in Severe NAFLD, there was a statistically significant increase in REM-C level at 3 months with a subsequent non statistically significant decrease at 6 months. Figure 3 displays the results for this analysis and it is possible to observe a positive association between increasing levels of REM-C and NAFLD severity.

## 4. Discussion

In this research, all diets had a direct effect in lowering the levels of REM-C after 3 and 6 months of intervention in subjects with MetS. In adherent subjects, this effect was stronger mostly among LGIMD as compared to the control group.

Besides, REM-C levels were higher in subjects with moderate or severe NAFLD than in mild or absent ones, suggesting an association between liver metabolic function and REM-C: for the duration of the intervention, the less severe the degree of NAFLD, the lower the level of REM-C. Furthermore, while in the whole sample, there was a statistically significant decreasing trend among absent NAFLD and moderate NAFLD subjects. In the adherent group, the behavior was different: there was a statistically significant decreasing trend among moderate NAFLD subjects and a statistically significant increase at 3 months with a subsequent non significative decrease in the severe NAFLD subjects. These contrasting findings could be attributed to the role of diet in liver metabolism [43]. It is possible to speculate that some subjects had a higher intake of saturated fatty acids (SFA), although adherent to the diet. As shown in Appendix B (Table A1), the mean intake of SFAs in the LGIMD reached 15% of total Kcal, while international guidelines recommend only 10% [44]. While greatly reducing the consumption of red and processed meat, the intake of SFAs and dietary cholesterol may have increased from the consumption of eggs, dairy products, and some fatty fish, such as mackerel. Although these products are typical of the Mediterranean diet, consumed in greater quantities, they may have worsened the NAFLD severity. There is evidence that SFAs reduce peripheral tissue sensitivity to insulin, while monounsaturated fatty acids (MUFA) can offset this effect [45,46].

A relationship between dietary cholesterol and cardiovascular mortality is widely documented [47,48,49]. In particular, among the diet-related factors, SFAs have the greatest impact on LDL-C. An increase by 0.8–1.6 mg/dL of LDL-C is estimated for each 1% increase in SFAs intake [50]. Partially hydrogenated fatty acids are the main source of fats in foods of industrial origin, ranging from 2% to 5% of daily food intake in western countries. Their effect on LDL-C values is similar to that of SFAs [51]. Excess alcohol consumption is associated with increased adipocyte lipolysis and flow of Free Fatty Acids (FFAs) to the liver, resulting in increased VLDL production [52]. The western diet, characterized by an overconsumption of fructose, red meat, alcohol, soft drinks, and SFAs, in addition to a reduced intake of dietary fiber and omega-3 rich foods, displays an unfavorable relationship with the occurrence of NAFLD. This association may be explained by different mechanisms and attributed to each of the components that characterize this dietary pattern.

Our results show that changes in lifestyle and, in particular, eating habits are of great importance for the prevention and clinical management of MetS and NAFLD [43].

The Mediterranean way of eating is recommended for the treatment of NAFLD thanks to its potential to improve metabolic alterations such as insulin resistance and lipid profile, even without any accompanying weight loss [26]. The LGIMD owes its positive effects to the high intake of MUFA from extra virgin olive oil, polyunsaturated fatty acids (PUFA) from fish and nuts, fruit fibers, legumes and vegetables, as well as the reduced intake SFA and hydrogenated fatty acids. A greater intake of PUFA omega-3 and MUFAs are considered as beneficial since they limit inflammation and endothelial dysfunction and improve dyslipidemia [53,54]. In addition, replacing packaged and processed food with fresh food leads to a reduction in salt intake. Salt intake has also been suggested to be associated with an increased risk for NAFLD [25].

Furthermore, the greater intake of fiber and whole grains in LGID and LGIMD may have influenced the glycemic index control and, as a consequence, an improvement in the IR condition in NAFLD rather than in the MD diet. Moreover, the use of whole grains in LGID and LGIMD was fundamental in inducing further benefits on the cardiometabolic risk profile as they contain phenolic compounds, which increase antioxidant and anti-inflammatory activities [55,56]. Whole grains have the potential to beneficially influence the gut microbiota composition [57], which can be relevant due to the importance of the gut-liver axis in the onset and progression of NAFLD [58].

Subjects with severe NAFLD had the most impaired metabolic profile and showed significantly higher levels of REM-C at baseline. Our results are consistent with recent literature. In a prospective observational study, Pastori et al. (2018) [21] found that NAFLD was independently associated with higher serum fasting REM-C. In the same work, high REM-C levels were predictive of major adverse cardiovascular and cerebrovascular events in patients with NAFLD. In fact, it is known that subjects with NAFLD have prolonged metabolic disorders involving the glycemic and lipid balance. Dyslipidemia occurs with increased triglycerides, elevated atherogenic lipoproteins (LDL-C and VLDL-C), and reduced HDL-C levels. A glycemic imbalance develops, instead, with the condition of greater resistance to insulin. In this context, it was recently shown that patients with hepatic IR exhibited a higher postprandial triglyceride response than patients with muscle IR or without IR [59]. In the insulin resistant state, an increased lipolysis of adipocytes causes an increased flow of FFAs in the liver, which feeds APO-B lipidation, leading to an increase in VLDL production and de novo liver lipogenesis [60], which is ultimately shown by an increase in remnant cholesterol levels. As suggested by other studies, REM-C is a promising tool for CVD risk assessment since REM-C shares with LDL-C has the potential to enter and be trapped in the arterial wall, triggering plaque formation and progression [61,62].

Some methodological issues need to be considered. The strength of this study is the setting from which the population came from. One limitation of our work stems from the fact that calculated REM-C may not be as accurate as direct measurement, but it is difficult to assess all remnants components as it is more expensive and time consuming [63]. Therefore, calculated REM-C is an alternative approach to the direct measurement of REM-C, which has already been used in the studies of large Danish cohorts [10,11,12,13,14,15,16]. The fact that the sample size was not established based on the levels of REM-C can be criticized, but this is a secondary analysis and regardless, the sample came from the general population, so it is reasonable to think that it represents the true distribution of this marker.

## 5. Conclusions

In conclusion, fasting REM-C level is independently associated with the grade of severity of NAFLD and the improvement of the metabolic state of the liver and IR induced by the diet reduces the REM-C level. The Mediterranean dietary pattern may be a useful strategy for reducing the degree of NAFLD in patients with MetS and high-risk CVD to decrease the residual risk associated with REM-C. The LGIMD diet has proven to be more effective than other diets in reducing REM-C in the total sample. However, it is essential to control the SFA intake, especially in subjects with moderate and severe NAFLD.

## Figures and Tables

**Figure 1 nutrients-12-01674-f001:**
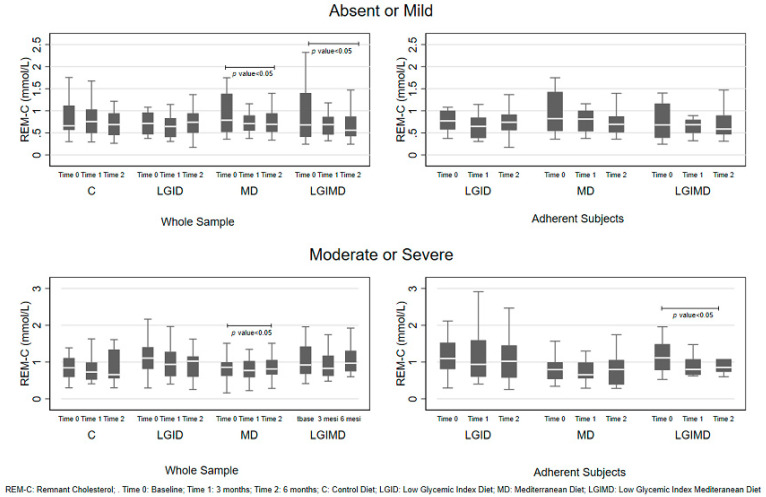
Remnant cholesterol level by non-alcoholic fatty liver disease severity, diet, and time.

**Figure 2 nutrients-12-01674-f002:**
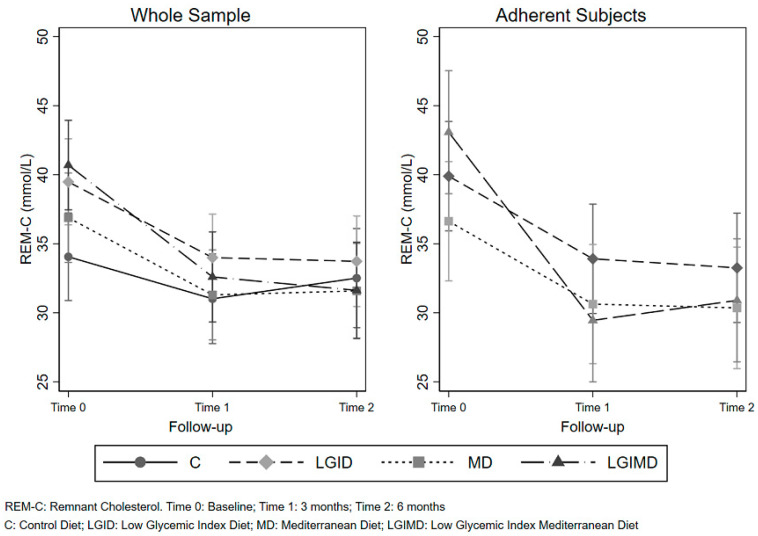
Repeated measures analysis of variance. Marginal effects of diet and time on remnant cholesterol.

**Figure 3 nutrients-12-01674-f003:**
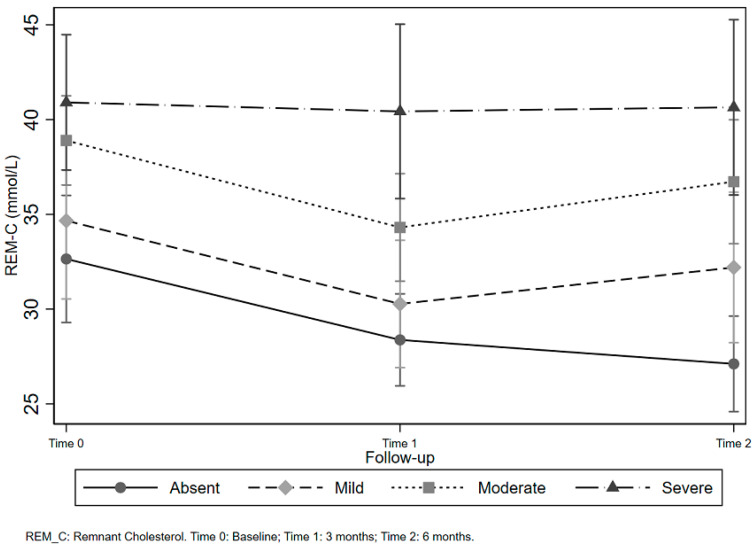
Remnant cholesterol levels by non-alcoholic fatty liver disease severity and time.

**Table 1 nutrients-12-01674-t001:** Descriptive statistics (mean (±standard deviation) or relative frequency (%)) of the main characteristics of the participants by intervention and time.

Variables *	Time	Diet			
		^1^ C	^2^ LGID	^3^ MD	^4^ LGIMD
		Mean (SD)	Mean (SD)	Mean (SD)	Mean (SD)
N		61	62	57	57
Age (years)		59.15 (12.76)	57.50 (10.44)	59.07 (10.83)	57.81 (9.77)
Gender					
Female		23 (38%)	25 (40%)	20 (35%)	25 (44%)
Male		38 (62%)	37 (60%)	37 (65%)	32 (56%)
^5^ REM-C (mmol/L)	Baseline	0.91 (0.56)	1.04 (0.49)	0.94 (0.57)	1.05 (0.60)
	3 months	0.81 (0.38)	0.87 (0.58)	0.80 (0.32)	0.84 (0.46)
	6 months	0.79 (0.40)	0.86 (0.47)	0.84 (0.39)	0.83 (0.45)
^6^ SBP (Hg mm)	Baseline	135.90 (16.75)	139.35 (17.33)	134.74 (13.38)	138.51 (17.58)
	3 months	129.91 (14.31)	128.00 (15.82)	127.32 (15.07)	127.50 (15.98)
	6 months	127.50 (11.35)	125.88 (15.92)	123.82 (15.02)	128.00 (17.14)
^7^ DBP (Hg mm)	Baseline	84.18 (9.97)	86.29 (6.83)	86.14 (7.50)	87.37 (7.97)
	3 months	76.49 (9.30)	76.58 (7.89)	77.32 (7.20)	79.02 (8.17)
	6 months	76.46 (8.05)	77.32 (9.39)	75.88 (7.73)	76.50 (9.11)
^8^ BMI	Baseline	33.19 (4.33)	33.42 (6.68)	33.05 (3.83)	34.92 (5.75)
	3 months	31.93 (4.20)	31.65 (6.12)	31.31 (3.61)	32.72 (5.40)
	6 months	31.15 (3.79)	31.41 (6.09)	31.01 (3.66)	31.98 (4.93)
^9^ WC (cm)	Baseline	106.70 (10.65)	105.94 (11.40)	104.78 (9.58)	109.28 (12.89)
	3 months	104.56 (10.02)	102.35 (10.89)	101.55 (9.34)	105.68 (12.78)
	6 months	104.85 (9.21)	102.04 (10.82)	101.36 (9.26)	104.51 (11.48)
^10^ TGL (mmol/L)	Baseline	1.99 (1.22)	2.26 (1.07)	2.06 (1.25)	2.29 (1.32)
	3 months	1.77 (0.83)	1.90 (1.27)	1.75 (0.70)	1.84 (1.00)
	6 months	1.73 (0.87)	1.88 (1.03)	1.83 (0.85)	1.80 (0.98)
^11^ TC (mmol/L)	Baseline	5.17 (1.13)	5.44 (1.07)	5.31 (1.08)	5.12 (1.06)
	3 months	4.98 (1.03)	5.04 (1.13)	5.01 (0.81)	5.00 (0.90)
	6 months	4.83 (0.91)	5.11 (1.14)	5.15 (0.78)	5.00 (0.94)
^12^ HDL-C (mmol/L)	Baseline	1.27 (0.33)	1.22 (0.25)	1.23 (0.24)	1.20 (0.33)
	3 months	1.34 (0.50)	1.24 (0.30)	1.28 (0.27)	1.22 (0.27)
	6 months	1.17 (0.31)	1.26 (0.23)	1.27 (0.26)	1.24 (0.30)
^13^ LDL-C (mmol/L)	Baseline	2.98 (0.86)	3.18 (0.89)	3.14 (0.82)	2.90 (0.95)
	3 months	2.82 (0.83)	2.93 (0.97)	2.92 (0.77)	2.94 (0.78)
	6 months	2.86 (0.72)	2.98 (0.94)	3.03 (0.82)	2.93 (0.87)
^14^ HOMA-IR	Baseline	3.94 (3.15)	4.53 (2.07)	4.31 (3.72)	4.99 (3.44)
	3 months	3.44 (2.61)	3.26 (1.67)	2.97 (1.53)	3.55 (2.54)
	6 months	3.07 (3.01)	3.49 (3.49)	3.10 (2.10)	3.64 (3.34)
^15^ VAT	Baseline	64.53 (17.86)	59.04 (21.02)	62.10 (17.98)	63.26 (16.96)
	3 months	56.58 (16.21)	53.97 (14.74)	54.70 (15.30)	57.03 (13.89)
	6 months	56.89 (17.20)	54.34 (13.71)	55.07 (14.85)	59.94 (18.42)
Glucose (mmol/L)	Baseline	6.27 (1.36)	6.89 (1.86)	6.39 (1.51)	7.11 (2.24)
	3 months	5.83 (1.18)	5.98 (0.89)	5.78 (1.25)	6.37 (1.67)
	6 months	5.75 (1.58)	5.90 (0.91)	5.75 (1.15)	6.36 (1.66)
^16^ APO B (g/L)	Baseline	1.02 (0.25)	1.12 (0.28)	1.09 (0.27)	1.06 (0.29)
	3 months	0.91 (0.27)	0.94 (0.31)	0.95 (0.26)	0.98 (0.29)
	6 months	0.94 (0.24)	0.94 (0.26)	0.94 (0.26)	0.94 (0.25)
^17^ HbA1c (mmol/mol)	Baseline	5.66 (0.56)	6.16 (1.09)	5.92 (0.90))	6.42 (1.42)
	3 months	5.67 (0.64)	5.85 (0.89)	5.69 (0.34)	5.98 (0.91)
	6 months	5.54 (0.56)	5.76 (0.59)	5.67 (0.56)	5.97 (0.84)
^18^ NCEP-ATP III criteria					
0–2	Baseline	19 (31%)	7 (11%)	12 (21%)	14 (25%)
	3 months	29 (48%)	25 (40%)	34 (60%)	23 (40%)
	6 months	28 (46%)	30 (48%)	30 (53%)	24 (42%)
3	Baseline	26 (43%)	22 (35%)	26 (46%)	13 (23%)
	3 months	21 (34%)	24 (39%)	13 (23%)	15 (26%)
	6 months	10 (16%)	16 (26%)	15 (26%)	14 (25%)
4	Baseline	12 (20%)	26 (42%)	18 (32%)	21 (37%)
	3 months	8 (13%)	12 (19%)	9 (16%)	13 (23%)
	6 months	22 (36%)	15 (24%)	12 (21%)	12 (21%)
5	Baseline	4 (7%)	7 (11%)	1 (2%)	9 (16%)
	3 months	3 (5%)	1 (2%)	1 (2%)	6 (11%)
	6 months	1 (2%)	1 (2%)	0 (0%)	7 (12%)

* No statistically significant difference among groups were found at baseline; ^1^ C: Control diet; ^2^ LGID: Low Glycemic Index Diet; ^3^ MD: Mediterranean Diet; ^4^ LGIMD: Low Glycemic Index Mediterranean Diet; ^5^ REM-C: Remnant Cholesterol; ^6^ SBP: Systolic Blood Pressure; ^7^ DBP: Diastolic Blood Pressure; ^8^ BMI: Body Mass Index; ^9^ WC: Waist Circumference; ^10^ TGL: Triglycerides; ^11^ TC: Total Cholesterol; ^12^ HDL-C: High-Density Lipoprotein Cholesterol; ^13^ LDL-C: Low-density Lipoprotein Cholesterol; ^14^ HOMA-IR: Homeostasis model assessment for insulin resistance; ^15^ VAT: Visceral Adiposity Thickness; ^16^ APO-B: Apolipoprotein B; ^17^ HbA1c: Glycated Hemoglobin; ^18^ NCEP-ATP III criteria: National Cholesterol Education Program (NCEP) Adult Treatment Panel (ATP) III criteria [42].

**Table 2 nutrients-12-01674-t002:** Remnant cholesterol levels by NAFLD severity and time in control subjects, whole sample, and adherent subjects.

NAFLD ^1^	Time	Control Diet		Intervention Subjects		Adherent Subjects	
		N (%)	Mean (SD)	N (%)	Mean (SD)	N (%)	Mean (SD)
Absent	Baseline	12 (23.53)	0.67 (0.39)	30 (21.28)	0.87 (0.54)	18 (20.93)	0.80 (0.52)
	3 months	22 (43.14)	0.80 (0.39)	54 (38.30)	0.75 (0.44)	30 (34.88)	0.71 (0.31)
	6 months	17 (33.33)	0.61 (0.23)	57 (40.43)	0.71 (0.31)	38 (44.19)	0.73 (0.29)
Mild	Baseline	14 (37.84)	0.94 (0.38)	30 (27.78)	0.95 (0.57)	20 (25.32)	0.93 (0.38)
	3 months	12 (32.43)	0.79 (0.32)	44 (40.74)	0.77 (0.46)	33 (41.77)	0.72 (0.30)
	6 months	11 (29.73)	0.92 (0.45)	34 (31.48)	0.76 (0.43)	26 (32.91)	0.71 (0.34) *
Moderate	Baseline	24 (45.28)	0.96 (0.71)	75 (44.91)	1.05 (0.50)	45 (45.45)	1.08 (0.56)
	3 months	15 (28.30	0.79 (0.44)	53 (31.74)	0.88 (0.44)	30 (30.30)	0.89 (0.53)
	6 months	14 (26.42)	0.82 (0.39)	39 (23.35)	0.98 (0.47)	24 (24.24)	0.90 (0.50)
Severe	Baseline	11 (44.0)	1.02 (0.52)	41 (46.07)	1.09 (0.63)	17 (48.57)	1.25 (0.81)
	3 months	8 (32.0)	0.90 (0.39)	21 (23.60)	1.10 (0.53)	7 (20.00)	1.37 (0.67)
	6 months	6 (24.0)	1.03 (0.54)	27 (30.34)	1.03 (0.51)	11 (31.43)	1.23 (0.65)

^1^ NAFLD: Non-Alcoholic Fatty Liver Disease. * Anova test *p* < 0.05.

**Table 3 nutrients-12-01674-t003:** Repeated measures analysis of variance. Principal effects of diets and time and contrast of time within each level of diet in the whole sample and adherent subjects.

Remnant Cholesterol	Whole Sample				Adherent Subjects			
	β	SE	*p*-Value	(CI 95%)	β	SE	*p*-Value	(CI 95%)
Diet								
^1^ C	Referent							
^2^ LGID	−13.91	10.27	0.176	(−34.11; 6.28)	Referent			
^3^ MD	−23.76	10.28	0.021	(−43.97; −3.56)	37.02	10.39	0.000	(16.52; 57.51)
^4^ LGIMD	−36.40	10.28	0.000	(−56.61; −16.20)	38.66	10.40	0.000	(18.15; 59.18)
Time								
Baseline	Referent				Referent			
3 months	−3.04	2.32	0.190	(−7.56; 1.51)	−5.98	2.84	0.036	(−11.58; −0.38)
6 months	−1.55	2.47	0.532	(−6.41; 3.32)	−6.64	2.84	0.020	(−12.24; −1.04)
Time #Diet:								
(3 Months vs. Baseline) # C	−3.04	2.32	0.190	(−7.59; 1.51)				
(6 Months vs. Baseline) # C	−1.55	2.47	0.532	(−6.41; 3.32)				
(3 Months vs. Baseline) # LGID	−5.49	2.26	0.016	(−9.92; −1.05)	−5.98	2.84	0.036	(−11.58; −0.38)
(6 Months vs. Baseline) # LGID	−5.75	2.32	0.013	(−10.31; −1.20)	−6.64	2.84	0.020	(−12.24; −1.04)
(3 Months vs. Baseline) # MD	−5.58	2.34	0.017	(−10.18; −0.99)	−6.00	3.09	0.054	(−12.10; 0.10)
(6 Months vs. Baseline) # MD	−5.30	2.42	0.029	(−10.06; −0.54)	−6.28	3.13	0.046	(−12.45; −0.10)
(3 Months vs. Baseline) # LGIMD	−8.10	2.34	0.001	(−12.69; −3.51)	−13.63	3.19	0.000	(−19.93; −7.32)
(6 Months vs. Baseline) # LGIMD	−9.07	2.44	0.000	(−13.87; −4.28)	−12.18	3.19	0.000	(−18.48; −5.88)

# by; ^1^ C: Control diet; ^2^ LGID: Low Glycemic Index Diet; ^3^ MD: Mediterranean Diet; ^4^ LGIMD: Low Glycemic Index Mediterranean Diet.

**Table 4 nutrients-12-01674-t004:** Repeated measures analysis of variance. Principal effects of NAFLD and time and contrast of time within each level of NAFLD in the whole sample and adherent subjects.

Remnant Cholesterol	Whole Sample				Adherent Subjects			
	β	SE	*p*-Value	(CI 95%)	β	SE	*p*-Value	(CI 95%)
^1^ NAFLD								
Absent	Referent				Referent			
Mild	−15.47	12.15	0.204	(−39.41; 8.46)	35.97	11.93	0.003	(12.27; 59.67)
Moderate	−5.89	14.04	0.675	(−33.56; 21.77)	41.27	13.04	0.002	(15.38; 67.15)
Severe	2.93	13.86	0.833	(−24.39; 30.24)	23.00	11.80	0.054	(−0.41; 46.42)
Time								
Baseline	Referent				Referent			
3 months	−4.27	2.16	0.049	(−8.52; −0.019)	−4.56	3.26	0.165	(−11.03; 1.91)
6 months	−5.53	2.24	0.014	(−9.95; −1.11)	−5.13	3.32	0.126	(−11.73; 1.46)
Time # NAFLD:								
(3 months vs. Baseline) Absent	−4.27	2.16	0.049	(−8.52; −0.019)	−4.56	3.26	0.165	(−11.03; 1.91)
(6 months vs. Baseline) Absent	−5.53	2.24	0.014	(−9.95; −1.11)	−5.13	3.32	0.126	(−11.73; 1.46)
(3 months vs. Baseline) Mild	−4.39	3.00	0.145	(−10.31; 1.52)	−5.46	3.80	0.154	(−13.01; 2.09)
(6 months vs. Baseline) Mild	−2.47	3.28	0.452	(−8.92; 3.99)	−3.15	3.90	0.421	(−10.91; 4.60)
(3 months vs. Baseline) Moderate	−4.59	2.06	0.027	(−8.66; −0.52)	−6.17	2.99	0.042	(−12.11; −0.23)
(6 months vs. Baseline) Moderate	−2.18	2.23	0.331	(−6.58; 2.23)	−6.80	3.14	0.033	(−13.02; −0.57)
(3 months vs. Baseline) Severe	−0.47	3.27	0.885	(−6.92; 5.97)	12.33	5.94	0.041	(0.53; 24.13)
(6 months vs. Baseline) Severe	−0.26	3.43	0.940	(−7.01; 6.49)	5.97	5.52	0.282	(−4.99; 16.94)

^1^ NAFLD: Non-Alcoholic Fatty Liver Disease; # by.

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
