# Peer review of "The Effect of Three Mediterranean Diets on Remnant Cholesterol and Non-Alcoholic Fatty Liver Disease: A Secondary Analysis"

_nutrients, 2020, doi:10.3390/nu12061674_

Round 1
Reviewer 1 Report
I have no other comments.
Reviewer 2 Report
This study aimed to evaluate the effects of different type of healthy diet (low glycemic (LG); Mediterranean (MD); and (LG + MD) on fasting remnant cholesterol from patients with metabolic syndrome and NAFLD. The REM-C is presented and evaluated as a maker of NAFLD. Clinical and biological analyses were processed for the diagnostic of NAFLD.
This study is a secondary analysis of the well-known MEDIDIET study. Sujet were randomly assigned to one of the three diet or to a control diet with usual guidelines to prevent CV disease. The diet was followed at 3 and 6 months
This study is well presented : the introduction and methodology well described.
The authors conclude to a more efficient LGMD diet than the two others. All the diet improve the le level of REM-C compared to control. There was a trend to a significant decrease with control. An unexpected increase was observed for adherent with severe NAFLD with the LGMD diet.
We would like that the three diet would be combined to improve the beneficial effect of diet compared to control.
This manuscript is a resubmission of an earlier submission. The following is a list of the peer review reports and author responses from that submission.
Round 1
Reviewer 1 Report
The study aims to analyse the effect of different diets on fasting remnant cholesterol (REM-C) levels in patients with metabolic syndrome (Mets) and the association between REM-C levels and Non- Alcoholic Fatty Liver Disease (NAFLD).
The aim of the article is original and interesting, however, I don´t feel the design is suitable for answering the question posed, as no accurate information regarding the concrete characteristic of each diet is included, which makes difficult to understand how a particular type of diet is associated with the outcome.
In general, the results are difficult to follow, it is difficult to identify the procedures followed with several aspects contributing to this confusion: the number of subjects in each group of adherence to diet is missing; they described results referring to table 4, and no table 4 is included in the manuscript; what the title of figure 2 means? (Figure 2. Expected Remnant Cholesterol level by adherence to diet…???); what is the significance of the control group?
With the paper lacking to show sufficient information regarding the data recorded about diet is not possible to understand the effect of a diet on any outcome, and difficult to have any conclusion supported by the results.
Reviewer 2 Report
1.The study design and result was related to the severe fatty liver , but it was not mention in the title.
2. How's the visceral adiposity thickness (VAT) acquired? Please further explain in the method section.
3. In Table 1 ,is there any significance difference among the variables of the 4 groups of diet? no p-value was shown. And same questions for Table 2, is there any difference of remnant cholesterol among the adherence and non adherence group from baseline to 3 months and 6 months?
4. Please define adherence and non adherence group. How are the authors monitor if the study group is adherence to the diet or not? What is the monitoring system of the authors toward the adherence group.Please further explain in the method section.
5. In table 3 and figure 2 , the significant value of remnant cholesterol on 6 months treatment of LGIMD in severe fatty liver should be compared with the baseline value instead of 3 months value , which is increased in unknown reason and reduced to similar values with baseline value. For me , this study has no significant finding, if this is the only significant finding for the authors.
6. In line 265, Table 4 is typographical error of table 3.
7. In the discussion part line 318, authors mentioned the whole grains in the diet has benefit on cardiometabolic profile, however whole grains was just a part of LGIMD, it can't represent the LGIMD study group.
8.In short, authors tried to explain the relationship between the lowering of remnant cholesterol in LGIMD can improve the cardiometabolic risk factor of the study group, especially in the cases of severe fatty liver , but the evidences unable to convince the reader.
Reviewer 3 Report
This study is a randomized clinical trial as a part of the MEDIDIET study. Here the aim was to evaluate the effect of 6 months of a healthy diet on metabolic syndrome (MetS) score as primary outcomes. Elevated fasting remnant control and Non Alcoholic Fatty Acid Liver was recorded to evaluate the impact of the diet. This is an interesting topic since so far, the impact of REM-C on MetS and Non Alcoholic Fatty Acid Liver Disease has not been reported and it might be involved in recurrent cardiovascular events.
The study has been focused on three diet intervention : Mediterranean diet (MD), Low Glycemic diet (LGID) and low glycemic and Glycemic Index Mediterranean (LGIMD).
The presentation of the results are not really reflecting the conclusion, and the authors should explain some major points.
Minor Comments - In the result part, the number of subjects for a control diet group is mentioned, but in the materials and methods there is no reference to a control diet group. This control diet group must be described. Why this group was not included in the table and figures ? - Line 198 “ Baseline REM-C were slightly different among group” : is this difference significant or not? The statistical analyses must be added in the legend of table1. - Thanks to add in all tables and figures the significant signs. - Figure 1: In the legend some information are missing (square, circle, diamond are not explained in the legend). - Line 265 : There is no table 4: this comment is addressed to table 3. Major Comments. - Why the three diet were not compared to the control diet since there is a control diet group (line 191). - In the materials and methods : the authors announced that MetS score and NAFLD score (measured by ultrasound) are the primary and secondary outcomes. However, the effect of the diet on these score after 6 months interventional diet are not presented in a table. This table must be added. - It would deserve for Table 2 to give the variation between baseline and 6 months for the parameters measured. It is not clear regarding the comment on line 209-211 that the results on REM-C are more improved with adherence diet than with non-adherence. Once again the statistical data must be added in the table. - Table 2 NADFL and match comment on Line 212-213: the slightly effect is not obvious, therefore authors must describe on which data they referred to make there conclusion. It is not clear with which diet the effect on REM-C depending of severity of NAFLD has been measured. Fig.1 and line 214-216 : according to the author there is no effect of the diet in absent or mild NAFLD but a trend to lower REM-C concentration : is this trend significant or not and even if it is not significant what is the p value for this data? Once again it is not obvious that the effect is more important in moderate and severe than in absent or mild: for example regarding the LGMID. The decrease between baseline and 6 months for the absent and mild NAFLD seems more important than the variation for the same period for the moderate to severe. The authors should discuss with more details the effect of each diet in the result part. - Figure 2 : according to the author there is a significant decrease at 6 months in the REM-C level : this significant difference is it between baseline or with 3 months? regarding the plot, it is not clear that baseline is different with the REM-C level at six months. The significant signs must be added in the figure. In the discussion the author must gives their hypothesis to explain a such increase at 3 months. - Line 305-307 : which data in the results show that LGMID improves significantly the NAFDL scores ? this is not clear on which data the comment is based. -- It would be appreciated and helpful if the authors could add a table of anthropometric values at at 6 months as they did for table 1( weight; BMI; Cholesterol, glycemia) - Discussion: line 300-301 : this conclusion does not seem related to a data in result part. The authors should explain with more details which significant result support their comment.